# Human Cytomegalovirus Protein Expression Is Correlated with Shorter Overall Survival in Breast Cancer Patients: A Cohort Study

**DOI:** 10.3390/v15030732

**Published:** 2023-03-11

**Authors:** Joel Touma, Mattia Russel Pantalone, Afsar Rahbar, Yan Liu, Katja Vetvik, Torill Sauer, Cecilia Söderberg-Naucler, Jürgen Geisler

**Affiliations:** 1Institute of Clinical Medicine, University of Oslo, Campus Akershus University Hospital (AHUS), 1478 Nordbyhagen, Norway; 2Department of Oncology, Akershus University Hospital (AHUS), 1478 Nordbyhagen, Norway; 3Department of Medicine, Solna, Microbial Pathogenesis Unit, Karolinska Institutet, 17164 Solna, Sweden; 4Department of Neurology, Karolinska University Hospital, 17177 Stockholm, Sweden; 5Department of Clinical Molecular Biology, University of Oslo, 0315 Oslo, Norway; 6Department of Clinical Molecular Biology (EpiGen), Akershus University Hospital (AHUS), 1478 Nordbyhagen, Norway; 7Department of Breast and Endocrine Surgery, Akershus University Hospital (AHUS), 1478 Nordbyhagen, Norway; 8Department of Pathology, Akershus University Hospital (AHUS), 1478 Nordbyhagen, Norway; 9Institute of Biomedicine, Unit for Infection and Immunology, MediCity Research Laboratory, Turku University, 20520 Turku, Finland

**Keywords:** breast cancer, human cytomegalovirus, HCMV, prognosis, overall survival

## Abstract

*Background:* Human cytomegalovirus (HCMV) is increasingly suggested to be involved in human carcinogenesis and onco-modulation due to its ability to contribute to all hallmarks of cancer. Growing evidence demonstrates a link between HCMV infection and various malignancies, including breast cancer, which incidence and mortality are still on the rise. The etiology of breast cancer remains mostly unclear, leaving 80% of breast cancer cases considered to be sporadic. Identifying novel risk- and prognostic factors for improved breast cancer treatment and increased survival rates, were the objectives of this study. *Methods:* Automated immunohistochemical staining results for HCMV proteins in 109 breast tumors and lymph node metastasis were correlated with clinical follow-up data (>10 years). Statistical analyses for median Overall Survival (OS) were performed. *Results:* Survival analyses revealed shorter median OS for patients with HCMV-IE positive tumors of 118.4 months compared to 202.4 months for HCMV-IE negative tumors. A higher number of HCMV-LA positive cells in the tumors was also associated with a shorter OS in patients (146.2 months vs. 151.5 months). *Conclusions:* Our findings suggest a link between HCMV-infections and breast cancer prognosis, which paves the way for potential novel clinical intervention and targeted therapy that may prolong the overall survival of selected patients with breast cancer.

## 1. Introduction

Breast cancer is the most commonly diagnosed cancer and accounts for 24.5% of all diagnosed malignancies worldwide [1]. It is the leading cause of cancer related deaths among women, and took 685 000 lives in 2020, which accounts for 15.5% of all cancer deaths globally [2]. In spite of great improvements in treating breast cancer, global data imply that the incidence and mortality is still on the rise [3]. The strongest prognostic factor in breast cancer has for many years been the presence or absence of lymph node involvement [4]. Risk stratification has however evolved further in recent decades, and today includes the biologic subtype of breast cancer, which has proven to be another valuable prognostic factor [5]. Some other risk factors for breast cancer are related to the onset and severity of the disease; unhealthy lifestyle and late age at first full-term pregnancy, sex, aging, early age at menarche, family history and gene mutations [6]. However, the disease is still considered sporadic, and these documented risk factors are absent in up to 80% of patients at the time of diagnosis [7]. Therefore, it is necessary to explore additional risk- and prognostic factors for these patients.

### 1.1. Human Onco-Viruses

The link between virus infections and cancer is well documented [8], and lately this link has been reported also in breast cancer [9,10,11,12]. The knowledge of viral oncogenesis has grown over the past years, and currently 12% of all cancers are estimated to be caused by so called onco-viruses [8]. Seven human viruses are currently classified as oncogenic; Epstein–Barr virus, Human papillomavirus, Kaposi sarcoma-associated herpesvirus 8, Hepatitis B and C viruses, Human T-lymphotrophic virus 1 and Merkel cell polyomavirus [13]. These onco-viruses target many of the same host signaling pathways to induce oncogenesis, which opens for actionable therapeutic opportunities in terms of prophylaxis, diagnosis and treatment [14]. In support of this statement, established vaccines and anti-virals have significantly reduced the incidence of virus-induced cancers over the past decades [14].

Infections that are caused by onco-viruses are common in the general population, but they rarely result in cancer without one or more additional factors being present, such as chronic inflammation, immunosuppression or environmental mutagens [15]. Within this frame of reference, the interest regarding the ubiquitous human herpes virus 5 (HHV-5), more commonly known as human cytomegalovirus (HCMV) has emerged, as there is a growing number of publications reporting the association between HCMV and a variety of solid malignancies, including breast cancer [10,11,12,16].

### 1.2. Human Cytomegalovirus

Human cytomegalovirus (HCMV) belongs to the β-herpesvirus family and is an opportunistic DNA virus [17]. Up to 90% of the world’s population is infected with HCMV, a virus considered to be harmless in healthy individuals [17]. However, a primary HCMV infection or re-activation from its latent state can cause significant morbidity and mortality in immunocompromised patients [18]. HCMV has a broad cell tropism and when reactivated, it has the potential to cause disease in almost every organ in the human body [19,20].

HCMV can be transmitted between individuals via close personal contact, sexual contact, blood, stem cell or organ transplants, and breast milk is considered to be primary routes of transmissions in infants. Following primary infection, HCMV establishes latency in myeloid lineage cells [19]. Another site for latent infection and viral reactivation is the breast epithelium [16]. In a study presented by Hamprecht et al., it was reported that 90% of HCMV seropositive women’s breast milk samples were positive for this virus, causing early primary infection of infants [21]. Thus, the breast is an important site of HCMV latency and reactivation.

HCMV reactivation is mediated by inflammatory cytokines [22,23,24], and requires activation of the major immediate-early promoter (MIEP) and production of immediate early (IE) proteins, which is followed by expression of early (E) and late (L) proteins [25]. The IE proteins are viral transcription factors that regulate viral and host gene expression, while L proteins are structural proteins that forms the virus particle [26].

### 1.3. HCMV in Cancer

The association between HCMV to human malignancies appears to be significant, as HCMV proteins and nucleic acids have been found in >90% of glioblastoma, neuroblastoma, medulloblastoma, prostate cancer, colon cancer, Hodgkin and Non-Hodgkin Lymphoma, ovarian cancer, and breast cancer [27,28,29,30,31,32,33,34]. HCMV proteins have also been found in >90% of lymph node and brain metastases of breast and colon cancer [35]. In sharp contrast, HCMV protein expression is rarely found in normal tissue around primary or metastatic tumors [32,35]. As this virus can affect many tumor promoting mechanisms, its presence in primary and metastatic tumors raises the question of its relevance in cancer.

Within the host, HCMV provides immunosuppressive functions that promotes viral persistence, enhanced survival of infected cells and increased viral load [36]. HCMV can also directly manipulate the immune responses, and fuel the development of genetic alterations within the cell. Relevant to cancer biology, HCMV immediate early (IE) gene products can interfere with key cellular pathways that causes chromosomal anomalies, DNA damage and disruption of the DNA repair mechanisms, that results in genetic instability and mutations in infected cells, which are all hallmarks of cancer cells [37,38,39,40,41]. Furthermore, HCMV proteins can induce inflammation and angiogenesis, block apoptotic pathways, inhibit tumor suppressor protein functions, dysregulate cell cycle progression, and alter cellular metabolism to the Warburg effect [42,43,44,45,46].

Given these abilities, HCMV exhibits essential features for the oncogenic properties required for a tumor-associated virus. Various publications from the last decade demonstrate that HCMV is able to induce all the hallmarks of cancer, and several investigators have suggested that HCMV should be added to the list of human onco-viruses [40,47,48].

## 2. Materials and Methods

Immunohistochemical (IHC) staining is the gold standard for HCMV diagnostics in humans. We have previously reported an optimized automated immunohistochemical staining method to identify HCMV immediate early (IE) and late antigen (LA), in breast tumor tissue and metastatic descents from primary tumors [49]. For this article, we used previously scored results from our research group and correlated them with demographics and clinical follow-up data (>10 years) and performed survival statistical analyses.

### 2.1. Patient Samples

The cohort used in this retrospective observational study consisted of 101 breast tumor samples and 8 lymph node metastases obtained from 109 breast cancer patients treated at Akershus University Hospital, Norway between 1996 and 2010. All surgical specimens had been concomitantly collected in a general breast cancer biobank with signed written informed consent provided by all patients prior to participation. All patients diagnosed with primary breast cancer, suitable for immediate surgery, could be enrolled. Approval of ethical permission obtained by the regional ethical committee of south-east Norway (No: 2014-895). The cohort consisted of patients whose clinical information over a minimum period of 10 years, and paraffin-embedded tumors were available. No active selection took place.

### 2.2. Immunohistochemical Staining

All samples evaluated in this study were formalin-fixed, paraffin-embedded (FFPE) and as mentioned before, previously analyzed by an automated IHC staining method that we recently established in our research laboratory for detection of HCMV proteins [49].

Briefly, the slides were processed in Dako autostainer link 48 platform and consecutive sections were stained with antibodies for HCMV-IE (MAB810R, Merck Millipore Burlington, MA, USA), HCMV-LA (MAB8127, Merck Millipore) and CK20 (IR777, Dako A/S Glostrup, Denmark), the latter serving as negative control. The detailed immunohistochemical staining protocol is described in a previous publication [49].

Each slide was evaluated and scored for HCMV IE and LA protein positivity by an experienced professor and senior consultant in pathology at Akershus University Hospital. All slides were scored according to the percentage of HCMV positive cells: 0; <1%, 1; 1–24%, 2; 25–49%, 3; 50–74%, and 4; >75%. In our previous methodology study, a total of 111 breast cancer samples were stained. Two samples were from recurrent breast cancer from the same patients that underwent surgery for a second time, and were therefore excluded from further analyzes, given they were not suitable for survival and correlation analyses.

### 2.3. Statistical Analyses

Data for median Overall Survival (OS) time after first diagnosis were collected into a database and analyzed for relation to HCMV protein expression. Survival data are presented in graphs as Kaplan-Meier estimates, calculated from the time of surgery. Relapse free survival was calculated separately for local recurrence and metastatic recurrence as time interval between surgery and diagnosis of recurrence. Correlation among categorical and ordinal (IHC gradings) and continuous variables was calculated by multivariable nonparametric Spearman analyses. Associations between binominal values (ER, PGR, HER2) and HCMV IE and LA staining defined as “low” (score 0–2 less than 50% tumor cells positive) and “high” (score 3–4 ≥ 51% tumor cells positive) were calculated by Fisher´s exact test. Correlation among the mentioned variables in relation to survival was calculated with multiple Cox regression analyses. Values are presented as mean and mean standard deviation. All statistical hypotheses were two-sided, and a significance level of 5%, *p* < 0.05 was considered significant. Graph Pad Prism (version 9.2) and SPSS (version 26.0) were used for statistical analyses.

## 3. Results

A TMA series of 109 BC cases was analyzed in this study to determine immunohistochemical data correlations between HCMV proteins and patient outcomes. Demographics, clinical and histopathological characteristics collected from patient records at Akershus University Hospital in Norway, are summarized in Table 1. All the patients included were women; the median age at the time of surgery was 60 years (range 34 to 92 years).

The cases were previously stained for HCMV IE and LA proteins [49]. The results are summarized in Table 2 and Figure 1 shows representative staining examples of HCMV IE and LA positive tumors, respectively.

Overall survival (OS) analysis revealed that patients whose tumors were positive for HCMV-IE at the time of diagnosis had a shorter median OS than patients with IE negative tumors (118.4 vs. 202.4 months, *p* = 0.04; Figure 2).

Patients whose tumors were positive for LA had a median OS of 147.2 months as compared to 152.1 months for patients whose tumors were LA negative, but this difference in survival time did not reach statistical significance (*p* = 0.12; Figure 3, panel a). Notably, a statistical significance was observed when the survival of patients whose tumors had a ≥25% cells positive for LA (scores 2, 3 and 4) were compared to patients with a tumor containing <25% LA positive cells (scores 0 and 1) with a median overall survival of 146.2 months vs. 151.5 months, respectively (*p* = 0.04; Figure 3, panel b).

We next performed Spearman correlation tests for the immunohistopathological and clinico-pathological variables. The analyses revealed that the expression of IE and LA positively correlated within the tissues (*p* < 0.03), suggesting that HCMV IE expression proceeded to late protein production in breast cancer cells in vivo. No correlation was found between viral protein expression and Grade, T stadium or N stadium. We also performed log-rank analyses for relapse free survival for both local relapses or presence of distant metastases, but no significant difference was found between patients whose tumors were IE positive or negative (*p* = 0.16), nor with low or high LA staining (*p* = 0.12) in this regard.

We further examined if viral IE or LA positivity were associated to binomial clinico-pathological variables. We tabled the number of IE and LA positive and negative cases according to local relapses or metastatic relapses (absence vs. presence), ER, PGR and HER2 (positive vs. negative) and performed a Fisher’s exact test. No statistical significance was found in any of the different groups analyzed. However, we noted that 5 out of 33 (15.2%) tumors from patients experiencing relapses were HCMV-IE positive, while only 4 out of 76 (5.3%) patients who did not relapse had HCMV-IE positive tumors. Furthermore, we did not find any differential expression of IE and LA in the different BC histological subgroups (infiltrating ductal BC, infiltrating lobular BC, medullary BC, mucinous BC and mixed type BC) or to the tumor subtypes (Luminal A, Luminal B, HER2 or TNBC).

Finally, multiple Cox regression analyses showed that relapse was the most significant variable negatively associated with OS (*p* < 0.001) followed by older age at surgery (*p* = 0.001), and higher tumor grade (*p* = 0.05). A higher LA immunohistochemical score trended towards an association with shorter OS, but a statistical significance was not reached (*p* = 0.83).

## 4. Discussion

Identification of novel therapeutic targets and prognostic/predictive markers, is a pivotal task in clinical oncology. Accumulating evidence links HCMV to various malignancies and high-risk HCMV strains have recently been identified specifically in breast cancer [50,51]. Here, we report that a higher HCMV protein load in human breast cancer at the time of diagnosis is correlated with a shorter overall survival. In particular, patients whose tumors were positive for HCMV-IE, although representing only 8.3% of cases in the cohort, had a significant shorter median overall survival (118.4 vs. 202.4 months, *p* = 0.04). HCMV-LA was more widely expressed and present in 76% of tumors, and a higher LA load in the tumors (≥25% positive cells) was also associated with shorter median OS (146.2 months vs. 151.5 months, *p* = 0.04). Although not statistically significant, we observed that patients whose tumors were HCMV-IE positive had 3 times higher risk of experiencing loco-regional and/or distant relapse. This observation may suggest that this group of patients may be more prone to relapse, however, a larger number of cases is needed to confirm this preliminary finding.

We previously published, in a smaller cohort, that patients with a low-grade HCMV-IE infection in primary colon and breast cancers have significantly longer time to brain metastasis and OS [35]. Other researchers have also reported an association between the risk of developing metastasis in patients with breast cancer who are seropositive for HCMV or HCMV DNA positive [52]. Furthermore, HCMV was correlated to a shorter relapse free survival in patients with breast cancer [53]. In this context, we believe that the relevance of our present study resides in the fact that we validated the association between OS and presence of HCMV proteins in the tumor tissues of patients with breast cancer in a larger and well characterized cohort. These clinically relevant results were obtained from histological analyses performed on tissue samples stained with an automated technique that could be easily applied by others for routine clinical use [49].

Considering the literature and our own previous experience using a manual IHC method [32,54], we expected to detect similar levels of HCMV-IE and LA in cancer tissue specimens. When we evaluated the staining results from the automated IHC staining method established in our laboratory [49] and used in this study, this was not the case. Spearman correlation analyses revealed that IE expression positively correlated with LA expression, concurring with the anticipation that HCMV-IE expression proceed to LA expression in breast cancer cells in vivo. However, there was a discrepancy between the percentage of IE expressing cells and LA expressing cells. Most likely this was dependent on technical differences in the staining protocols, which resulted in loss of cytoplasmic IE protein expression when eliminating background staining for the automatic procedure. Thus, the samples with the strongest immunoreactivity remained IE positive and this staining result correlated with patient outcome. However, we cannot exclude the possibility that low IE reactivity exists in a larger percentage of breast cancer tumor specimens, but this was under the detection limit with the automated method.

We have previously reported that HCMV presence in human breast cancer is correlated to expression of the inflammatory proteins COX-2 and 5-LO and that HCMV is able to induce expression of these inflammatory proteins in breast cancer cell lines in vitro [54]. This feature of HCMV could promote a more aggressive breast cancer phenotype. The role of HCMV as a tumor promoting virus may depend on the viral strain carried by individual patients. Of high concern, certain clinical strains isolated from patients with breast cancer are able to induce oncogenic and stemness signatures in vitro [55,56,57]. In our study, HCMV positivity did not correlate statistically with other factors such as subtypes, grading, sex or age. Although previous publications indicate that HCMV may be associated predominantly with TNBC [58], higher tumor grade and invasive phenotype [59,60], we did not find a correlation between IE or LA staining and time to tumor relapse, or to presence of local or metastatic tumors, or breast cancer phenotype. This may be explained by the cohort size only containing 15 patients with TNBC. This breast cancer phenotype is notoriously resistant to therapy and has worse prognosis than other breast cancer types and has higher mortality rates due to the lack of efficient therapies and further larger studies are warranted to prove or disprove an association between TNBC and HCMV.

A correlation between HCMV presence in tumors and survival was first observed in patients with glioblastoma [61]. Several studies then indicated that HCMV is associated with higher malignancy grade and poor outcome not only in patients with glioblastoma and but also in patients with breast, colon, ovarian and prostate cancer [62,63,64,65,66]. If the virus plays a role as a tumor promoting agent for various cancer forms, viral protein expression should be considered and evaluated as a prognostic marker, and this could open for new therapeutic possibilities for patients.

Retrospective studies have shown that antiviral therapy against HCMV is associated with increased survival time in patients with glioblastoma [67,68,69]. A prospective study is currently ongoing at Karolinska University Hospital in Stockholm, Sweden (NCT04116411) to assess whether the antiviral drug valganciclovir can prolong survival of glioblastoma patients. Furthermore, pp65 mRNA dendritic cell vaccination also showed highly prolonged median overall survival in 11 patients diagnosed with glioblastoma, which further implies that control of HCMV in patients with glioblastoma may give patients a better prognosis [70]. The latter strategy is also under evaluation in a clinical trial (NCT03927222).

## 5. Conclusions

In this study, we found that higher HCMV protein expression in breast cancer biopsies at the time of primary diagnosis is significantly associated with a shorter overall survival in patients with breast cancer. Our data provides an important clinical validation to the numerous in vitro and in vivo studies demonstrating that HCMV could play a role in breast cancer pathogenesis [11,54,55,56,57,58,71,72]. These observations were derived from histological analyses performed on a cohort of tissue stained with an automated technique that could be easily applied by others for routine clinical use [49]. Thus, HCMV protein expression levels could potentially represent new prognostic markers for breast cancer.

Altogether, our findings are coherent with the previous literature regarding a putative role of HCMV in breast cancer pathogenesis and suggest that HCMV plays a potential role in promoting disease progression and clinical relapses in subgroups of patients with breast cancer. Therefore, antiviral treatment might represent a valid additional therapeutic option for selected patients and should be evaluated in future clinical trials.

## 6. Patents

C.S.-N. holds a patent for diagnostics and treatment of an HCMV variant strain found in cancer.

## Figures and Tables

**Figure 1 viruses-15-00732-f001:**
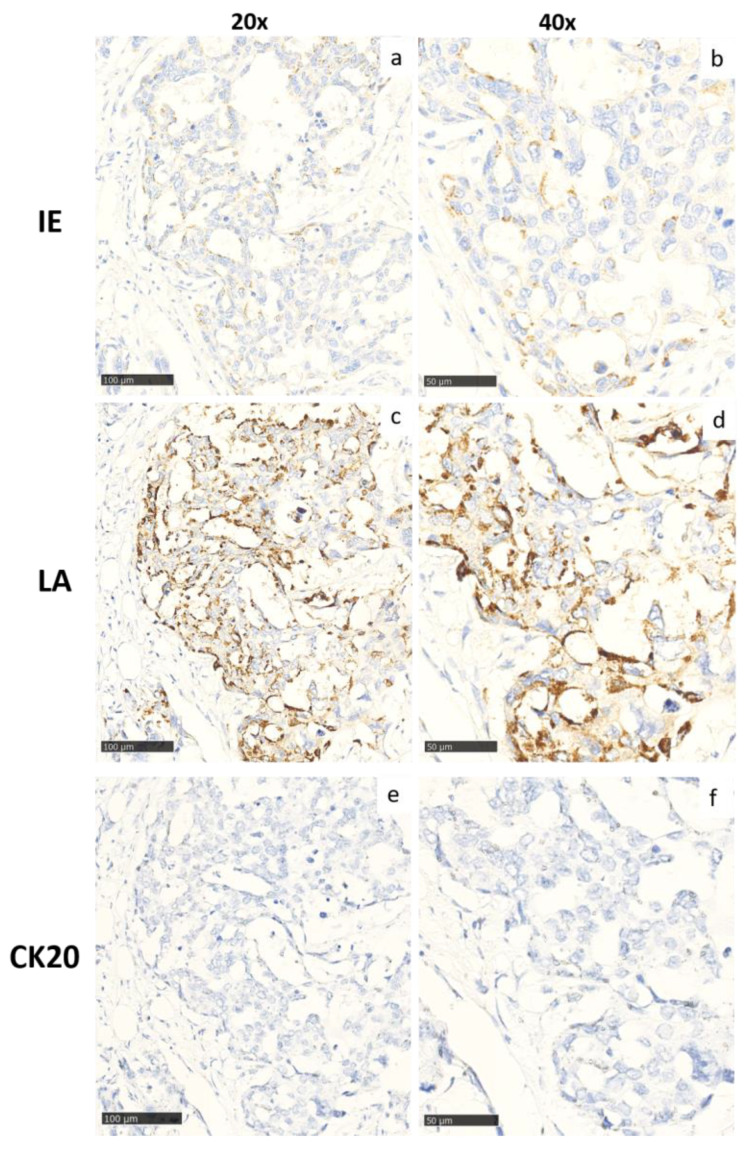
Representative images at two different magnifications (20x; (**a**,**c**,**e**) and 40x; (**b**,**d**,**f**)) of the automated immunostaining for HCMV-IE (**a**,**b**) and HCMV-LA (**c**,**d**) and Cytokeratin-20 (CK-20; (**e**,**f**) used as a control) in breast cancer. Cytoplasmic staining is observed for both HCMV proteins. In this analysis, LA was more widely and strongly expressed, while IE was more selectively expressed.

**Figure 2 viruses-15-00732-f002:**
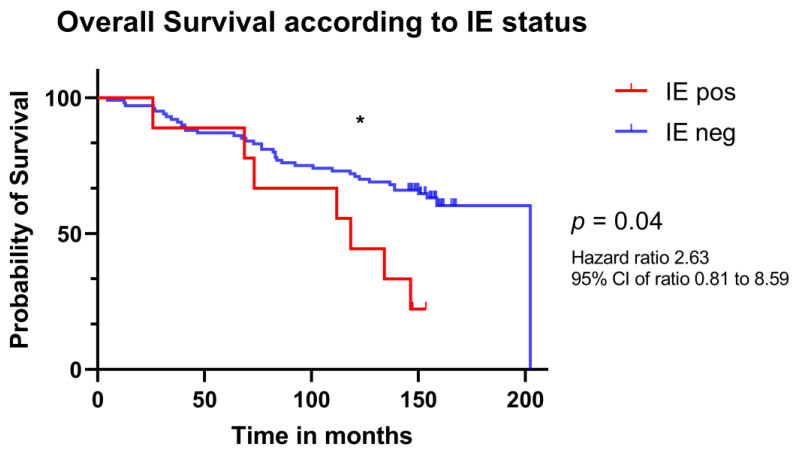
Kaplan–Meier estimates of overall survival in patients with breast cancer according to HCMV IE status at time of diagnosis. Estimated survival time for 9 patients whose tumors were IE positive (red) and of 100 patients whose tumors were IE negative (blue). Significance is indicated as following: * (*p* < 0.05).

**Figure 3 viruses-15-00732-f003:**
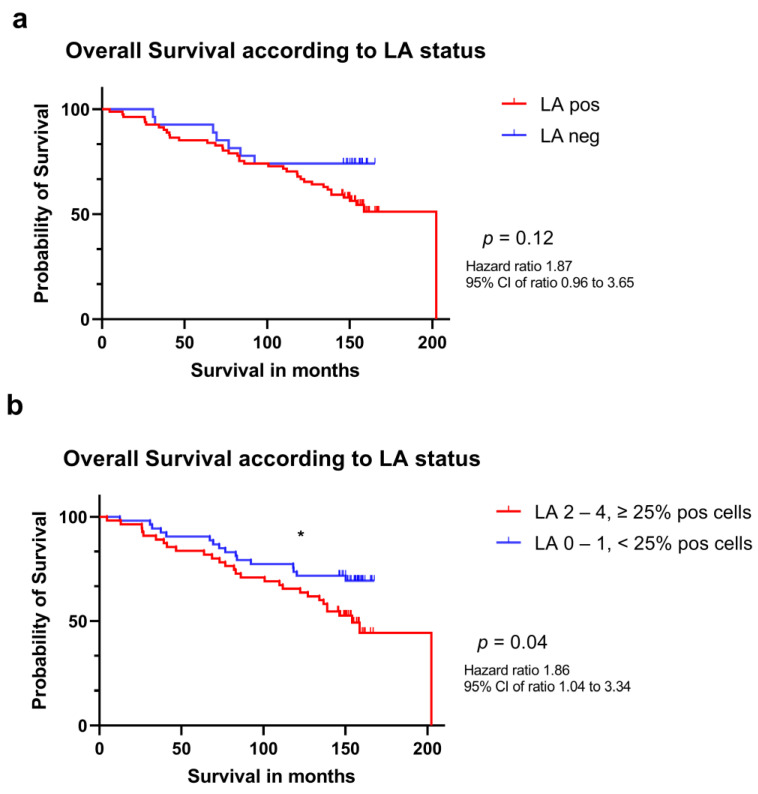
Kaplan-Meier estimates of survival in patients with breast cancer according to HCMV LA status. (**a**) estimated survival of 82 patients whose tumors were LA positive (red) and of 26 patients whose tumors were LA negative (blue). (**b**) estimated survival of 55 patients whose tumors showed ≥ 25% of LA positive cells and of 53 patients whose tumors showed less than 25% positive cells or were negative (blue). Significance is indicated as following: * (*p* < 0.05).

**Table 1 viruses-15-00732-t001:** Characteristics of the series of patients with breast cancer.

	N	(%)
Patients included	109	100
Median age at surgery (Y)	60 (34–92)	
Gender		
Females	109	(100)
Males	0	(0)
Histologicial grade (G)		
1	7	(6.4)
2	56	(51.4)
3	46	(42.2)
T ^1^ stadium		
1	52	(47.7)
2	53	(48.6)
3	4	(3.7)
N ^2^ stadium		
0	62	(56.9)
1	27	(24.8)
2	11	(10.1)
3	9	(8.2)
M ^3^ stadium		
0	108	(99.1)
1	1	(0.9)
Relapse occurence		
Loco-regional	10	(9.2)
Distant metastasis	33	(30.3)
Histology		
Infiltrating Ductal	90	(82.6)
Infiltrating Lobular	10	(9.2)
Medullary	1	(0.9)
Several types	8	(7.3)
Molecular subtype		
Luminal A	60	(55)
Luminal B	10	(9.2)
HER2	24	(22)
TNBC	15	(13.8)
HER2 ^4^		
Positive	26	(23.9)
Negative	83	(76.1)
PGR ^5^		
Positive	50	(45.9)
Negative	59	(54.1)
ER ^6^		
Positive	82	(75.2)
Negative	27	(24.8)

^1^ tumor size; T1 < 20 mm, T2 20–50 mm, T3 > 50 mm, ^2^ nodal involvement; N0 no cancer cells are seen in loco-regional lymph nodes, N1 cancer has spread to 1–3 axillary lymph nodes and/or internal mammary lymph nodes, N2 cancer has spread to 4–9 axillary lymph nodes or cancer has enlarged the internal mammary lymph nodes, N3 cancer has spread to 10 or more axillary lymph nodes or to clavicular lymph nodes, ^3^ metastases; M0 no distant spreading, M1 cancer has spread to distant organs, ^4^ human epidermal growth factor receptor 2 expression; positive when IHC 3+ or IHC 2+ and amplified, ^5^ progesterone receptor expression; positive > 10%, ^6^ estrogen receptor expression; positive > 1%.

**Table 2 viruses-15-00732-t002:** Scores of immunohistochemical staining for HCMV IE and HCMV LA, in the breast cancer samples.

Score	0		1		2		3		4		Total	
	*n*	%	*n*	%	*n*	%	*n*	%	*n*	%	*n*	%
HCMV IE	100	91.7	4	3.7	4	3.7	1	0.9	0	0	109	100
HCMV LA	26	24	27	25	22	20.2	17	15.6	16	14.7	108	100

Negative (0% of stained cells); score 1 (1–24% of stained cells); score 2 (25–49% of stained cells); score 3 (50–75% of stained cells); score 4 (> 75% of stained cells).

## Data Availability

All data is contained within the article, for further information please contact the corresponding authors.

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
