# Peer review of "Human Cytomegalovirus Protein Expression Is Correlated with Shorter Overall Survival in Breast Cancer Patients: A Cohort Study"

_viruses, 2023, doi:10.3390/v15030732_

Round 1

Reviewer 1 Report

The manuscript by Touma et al. is clearly written and adresses an important concern: the role of HCMV in oncomodulation/oncogenesis. The data are clearly presented but several concerns have to be addressed:

Major points:

1- The authors make the statement related to data from Figure 1 and Table 2: "LA was more widely and strongly expressed, while IE was more selectively expressed". Since a complete lytic viral cycle involves first IE antigens expression followed by E antigens and LA antigens expression we would expect to detect similar levels of IE and LA antigens in the study. This is clearly not the case. How can the authors explain this discrepancy?

In addition for Figure 1 negative controls have to be added.

2- Lanes 283-284: "Of high importance and concern, clinical strains isolated from patients with breast cancer are able to induce oncogenic and stemness signatures in vitro [61], and breast cancer derived HCMV strains have induced oncogenic transformation of normal mammary cells and given rise to tumors in mice models [62]". This sentence is misleading and not accurate.

It has never been shown that breast cancer derived HCMV strains have given rise to tumors in mice models. The second part of the sentence has to be deleted.

In addition to Ref 61, the paper by Nehme et al. has to be cited: eBioMedicine 2022, 80:104056 (doi: 10.1016/j.ebiom.2022.104056).

3- The paradigm of high-risk oncogenic HCMV strains has been recently highlighted to explain the role of HCMV in oncomodulation/oncogenesis. This issue has definitively to be addressed in the discussion section and the following article cited: Viruses. 2022 Nov 7;14(11):2462. doi: 10.3390/v14112462.

Author Response

Point-by-point response:

Response to comments provided by Reviewer 1:

The manuscript by Touma et al. is clearly written and addresses an important concern: the role of HCMV in oncomodulation/oncogenesis. The data are clearly presented but several concerns have to be addressed:

Point 1: The authors make the statement related to data from Figure 1 and Table 2: "LA was more widely and strongly expressed, while IE was more selectively expressed". Since a complete lytic viral cycle involves first IE antigens expression followed by E antigens and LA antigens expression we would expect to detect similar levels of IE and LA antigens in the study. This is clearly not the case. How can the authors explain this discrepancy?

In addition, for Figure 1 negative controls have to be added.

Response 1: Thank you for your positive and valuable feedback. We agree that this discrepancy should be addressed for clarification and hope you will find our addition in the discussion part, lines 340-349 (page 9), satisfactory. We believe indeed the paper has been enhanced with this suggested explanation.

We added Cytokeratin-20 staining as control in Figure 1 according to the reviewer´s suggestion.

Point 2: Lanes 283-284: "Of high importance and concern, clinical strains isolated from patients with breast cancer are able to induce oncogenic and stemness signatures in vitro [61], and breast cancer derived HCMV strains have induced oncogenic transformation of normal mammary cells and given rise to tumors in mice models [62]". This sentence is misleading and not accurate.

It has never been shown that breast cancer derived HCMV strains have given rise to tumors in mice models. The second part of the sentence has to be deleted.

In addition to Ref 61, the paper by Nehme et al. has to be cited: eBioMedicine 2022, 80:104056 (doi: 10.1016/j.ebiom.2022.104056).

Response 2: Thank you for your thoughtfulness by addressing this error that is now corrected in line 374 (page 10) under section 4. Also, we found the paper suggested by the reviewer to be cited, to add strength to line 369-371 (page 10) under section 4.

Point 3: The paradigm of high-risk oncogenic HCMV strains has been recently highlighted to explain the role of HCMV in oncomodulation/oncogenesis. This issue has definitively to be addressed in the discussion section and the following article cited: Viruses. 2022 Nov 7;14(11):2462. doi: 10.3390/v14112462.

Response 3: Thank you for drawing our attention this important paper within the field, we have now gladly cited it on line 326 (page 9) under the section 4.

Reviewer 2 Report

The manuscript entitled “Human Cytomegalovirus as a Prognostic Factor in Breast Cancer” seems to be a good research work. However, it lacks proper presentation. According it needs major modifications as suggested below.

1. The generalized title is not reflecting the type of study. The title needs modification based on the objectives and study type.

2. Almost half of the abstract is covered with the background. Results have been mentioned in one line. The abstract must be rewritten mentioning important outcomes of the result.

3. Keywords: Remove “OS” after “overall survival”.

4. The introduction part is too long with 53 references. The introduction needs to be concise and other relevant parts may be discussed in the discussion section (if needed).

5. The authors must proofread the whole manuscript with Grammarly or other English software for spelling/spacing etc.

6. The inclusion and exclusion criteria are missing in the material and method section.

7. The sample size is small in this study. Similar studies have been conducted in the past (lines 298-299). In such cases, the authors must compare their results with the most closely related studies to prove the importance of their study and new findings.

Author Response

Response to comments provided by Reviewer 2:

The manuscript entitled “Human Cytomegalovirus as a Prognostic Factor in Breast Cancer” seems to be a good research work. However, it lacks proper presentation. According it needs major modifications as suggested below

Point 1: The generalized title is not reflecting the type of study. The title needs modification based on the objectives and study type.

Response 1:  Thank you for your valuable observation, we have modified the title accordingly and hope it reflects the objectives and the type of the study more: “Human Cytomegalovirus Proteins are Correlated With Shorter Overall Survival in Breast Cancer Patients: a Cohort Study”.

Point 2: Almost half of the abstract is covered with the background. Results have been mentioned in one line. The abstract must be rewritten mentioning important outcomes of the result.

Response 2: We absolutely agree, the background part of the abstract has now been down-scaled and we have added relevant result outcomes in lines 35-37 (page 1).

Point 3: Keywords: Remove “OS” after “overall survival”.

Response 3: This modification has been done as suggested by the reviewer.

Point 4: The introduction part is too long with 53 references. The introduction needs to be concise and other relevant parts may be discussed in the discussion section (if needed).

Response 4: We do agree that the introduction should be concise and have now shortened the introduction although, we believe that it is important to maintain the appropriate references to give a comprehensive outlook of the complex problem of CMV in cancer to the readers.

Point 5: The authors must proofread the whole manuscript with Grammarly or other English software for spelling/spacing etc.

Response 5: This has been done prior to re-submission.

Point 6: The inclusion and exclusion criteria are missing in the material and method section.

Response 6: We have now specified the inclusion and exclusion criteria in the Materials and Methods section at lines 188-190 and 191-193 (page 3) as suggested by the reviewer.

Point 7: The sample size is small in this study. Similar studies have been conducted in the past (lines 298-299). In such cases, the authors must compare their results with the most closely related studies to prove the importance of their study and new findings.

Response 7: You are absolutely correct, the clinical relevance of our contribution for future implications is now further highlighted in lines 374-379 (page 10) section 4.

Round 2

Reviewer 1 Report

No additional comments.

Reviewer 2 Report

The authors have addressed the comments. The manuscript can be accepted for publication.